# Non-Coagulant Spinning of High-Strength Fibers from Homopolymer Polyacrylonitrile Synthesized via Anionic Polymerisation

**DOI:** 10.3390/polym16091185

**Published:** 2024-04-23

**Authors:** Ivan Yu. Skvortsov, Mikhail S. Kuzin, Pavel S. Gerasimenko, Maria V. Mironova, Yaroslav V. Golubev, Valery G. Kulichikhin

**Affiliations:** A.V. Topchiev Institute of Petrochemical Synthesis Russian Academy of Sciences, 119991 Moscow, Russia; kuzms@ips.ac.ru (M.S.K.); gerasimenko11507@yandex.ru (P.S.G.); mvmironova@ips.ac.ru (M.V.M.); yagolubev@gmail.com (Y.V.G.); klch@ips.ac.ru (V.G.K.)

**Keywords:** polyacrylonitrile, anionic polymerization, fibers, thermo oxidative stabilization, rheology, thermolysis, mechanotropic spinning, coagulant-free spinning, homopolymer

## Abstract

The rheological properties, spinnability, and thermal–oxidative stabilization of high-molecular-weight linear polyacrylonitrile (PAN) homopolymers (molecular weights *M*_η_ = 90–500 kg/mol), synthesized via a novel metal-free anionic polymerization method, were investigated to reduce coagulant use, enable solvent recycling, and increase the carbon yield of the resulting carbon fibers. This approach enabled the application of the mechanotropic (non-coagulating) spinning method for homopolymer PAN solutions in a wide range of molecular weights and demonstrated the possibility of achieving a high degree of fiber orientation and reasonable mechanical properties. Rheological analysis revealed a significant increase in solution elasticity (G′) with increasing molecular weight, facilitating the choice of optimal deformation rates for effective chain stretching prior to strain-induced phase separation during the eco-friendly spinning of concentrated solutions without using coagulation baths. The possibility of collecting ~80 wt% of the solvent at the first stage of spinning from the as-spun fibers was shown. Transparent, defect-free fibers with a tensile strength of up to 800 MPa and elongation at break of about 20% were spun. Thermal treatment up to 1500 °C yielded carbon fibers with a carbon residue of ~50 wt%, in contrast to ~35 wt% for industrial radically polymerized PAN carbonized under the same conditions.

## 1. Introduction

Polyacrylonitrile (PAN) is one of the most important industrial polymers. Synthetic fibers obtained from PAN and its copolymers have a set of properties, including resistance to main organic solvents, high mechanical strength, and many others [1,2]. First of all, PAN staples are used as a wool-like material and found a major application in mixing with natural wool. In addition, membrane materials based on PAN have become widespread. Such materials are resistant to fats and oil products and are used in various industries, in particular, for wastewater treatment [3,4]. Moreover, PAN has the greatest practical importance as a precursor of carbon fibers. This is due to the high specific strength and fiber stiffness, as well as the high carbon yield during carbonization [2,5]. In this case, the properties of the final product largely depend on the chemical structure and molecular weight of the initial polymer, which is determined primarily by the synthesis conditions.

Traditionally, PAN is synthesized by free radical polymerization in an emulsion, suspension, or solution. As a result, atactic polymers with a wide molecular weight distribution and a low degree of branching are prepared. The molecular weight of such polymers can reach over 500 kg/mol [6]. Controlled radical polymerization, in particular atom transfer radical and reversible addition−fragmentation chain-transfer polymerization [7,8,9,10], can provide a lower polydispersity index; however, the molecular weight of the synthesized polymers is significantly reduced, which negatively affects the mechanical properties of fibers.

An alternative solution to the problem of obtaining high-molecular-weight PAN can be anionic polymerization methods [11,12]. Anionic polymerization is interesting primarily because of the absence of chain termination and chain-transfer reactions. The synthesis of PAN by the anionic mechanism is characterized by a high yield of polymers, the possibility of carrying out the reaction at high rates, and control of the polymer molecular weight distribution [2,13,14]. The initiators of the anionic polymerization of acrylonitrile are alkali metals, as well as compounds based on them, such as alcoholates, amides, and hydroxides [14,15]. Due to the presence of highly polar –C≡N groups in PAN, the reaction proceeds homogeneously only in highly polar solvents, such as dimethyl sulfoxide (DMSO) [16,17], dimethyl formamide [13], and dimethyl acetamide [18]. Unfortunately, the majority of currently known anionic polymerization initiators exhibit instability in these solvents.

Estrin’s group proposed a method for the anionic polymerization of acrylonitrile using initiating systems based on several cyclic tertiary amines—1,4-diazabicyclo [2.2.2]octane (DABCO), 1,8-diazabicyclo-[5.4.0]-undecene-7 (DBU) and 1,5-diazabicyclo-[4.3.0]-nonene-5 (DBN)—in combination with lower olefin oxides (ethylene oxide, propylene oxide) [19,20,21]. The proposed initiating systems contain only atoms which are in the elemental coincidence with PAN composition: hydrogen, carbon, oxygen, and nitrogen. This eliminates the possibility of the presence of metal, phosphorus, or sulfur impurities in the polymer, which could degrade the mechanical characteristics of the final product [22]. Previously, this method was used to obtain and describe hyperbranched polymers [23], which performed well only in mixtures with linear PAN from the point of view of obtaining fibers and films with developed porosity and were not suitable for obtaining high-strength fibers.

Typically, to produce carbon fibers, PAN ternary copolymers containing carboxy groups are used to expand the exothermic peak during the thermal–oxidative stabilization process when the cyclization of nitrile groups is being [2,24,25,26,27] and alkyl acrylates to increase spinnability, because PAN homopolymers are more prone to gelation due to nitrile–nitrile interactions, which prevent chains from orienting during stretching [16,28,29]. However, such monomers do not participate in the formation of the polyconjugated system at this stage, and their presence can contribute to the uneven cyclization of PAN and the formation of defects [30,31]. That is why finding the conditions for spinning highly oriented fibers from PAN homopolymers does not lose its relevance, and here, the choice of fiber-spinning method is of particular importance.

The most common methods for producing precursor fibers are wet and dry–wet spinning [9,10,11,12,13,14], in which the formation of solid fibers occurs due to phase separation due to the interaction of a solution jet with a coagulant. To obtain defect-free fibers in this way and minimize the formation of a core–shell structure, multicomponent coagulants are often used to ensure smooth leaching of the solvent from the formed gel fiber [21,32,33]. An alternative is the recently developed mechanotropic spinning method, in which phase separation of the solution occurs because of strong uniaxial deformation [34]. In this case, the phase separation passes through the entire volume of the jet when a critical degree of deformation is reached, and the solvent migrates to the surface of the formed gel fiber. This process produces round fibers with a uniform transverse morphology. The mechanotropic spinning method was previously used for a series of PAN copolymers and showed the possibility of obtaining high-strength fibers in the presence of various alkyl acrylate monomers [5,35], but the question of the possibility of obtaining oriented and high-strength fibers from PAN homopolymers by this method remained open.

This work presents data on rheological studies of concentrated solutions of a series of high-molecular-weight linear homopolymers of acrylonitrile obtained by anionic polymerization with an initiating system based on tertiary amines in DMSO [21,36,37], a fiber-spinning procedure from these solutions, the measurement of the mechanical properties of fibers, the analysis of the thermal properties of the fibers during stabilization in air, and the subsequent carbonization in an inert atmosphere when heated to 1500 °C.

## 2. Materials and Methods

### 2.1. Materials

A series of polymers were synthesized using the anionic polymerization method. The synthesis was described in detail in [36]. Table 1 shows the main properties of the polymers used. As a comparison sample, the industrial homopolymer AN-C produced by Sigma Aldrich 181315 (Sigma-Aldrich Headquarters: St. Louis, MO, USA) with an *M_w_* of 150 kg·mol^−1^ was used.

### 2.2. Methods

#### 2.2.1. Preparation of Solutions

Polyacrylonitrile powder and DMSO of 99.5% purity from Ekos-1 (Ekos-1, Moscow, Russia) were mixed in glass vials with hermetically sealed lids. Different dissolution methods were employed based on the polymer concentration:-For solutions with a polymer concentration of 1–5 wt%, the mixture was stirred for 24 h at 50 °C using a magnetic stirrer. For concentrations below 1 wt%, the solution was sequentially diluted inside a Ubbelohde viscometer at 25 °C to determine the intrinsic viscosity.-Highly viscous solutions with a polymer content greater than 5 wt% were prepared using a paddle mixer with a J-shaped rotor. Mixing was conducted at 60 rpm for 24 h at 70 °C.-High-viscosity spinning solutions with concentrations above 20 wt% were prepared using a rotor speed of 10 rpm for 72 h at 70 °C.

#### 2.2.2. Rheology of Solutions

The solutions’ rheological behavior was studied using the rotational rheometer HAAKE MARS 60 (Thermo Scientific, Karlsruhe, Germany), with a cone–plate geometry, a cone diameter of 20 mm, and an angle of 1° between the cone and plate. The tests were carried out in a controlled shear rate mode in the range of 10^−3^–10^3^ s^−1^, as well as in a dynamic mode in the linear viscoelastic region (frequency range 0.1–628 rad/s, strain 1%). The measurement temperature was 70 °C.

#### 2.2.3. Fiber Spinning

The fibers were spun using the non-coagulant mechanotropic method [39] on a specialized spinning line at a relative air humidity of 20% in a special sealed set and a temperature of 25 °C. This technique involves stretching a highly viscous solution jet in the air, inducing substantial elongation ratios that trigger phase separation. During this process, the solvent is released from the center of the jet to the surface of the spinning fiber. When using environmental air with high humidity, two potential mechanisms of polymer separation from solution jets may occur: diffusion of air moisture into the jet, leading to coagulation of the solution, and strong extension, leading to phase separation with the solvent release.

To supply the solution, a Malvern Rosand RH10 capillary rheometer (Malvern, Malvern, UK) was used with a controlled solution flow rate. A monofilament spinneret with a 500 μm diameter was used. The spinning fiber underwent two stages of orientation stretching: initially in the air, followed by a water wash to eliminate any residual solvent on the fiber surface. Subsequently, the fibers were dried at 90 °C and underwent a final process of thermal drawing at 130 °C to orient the as-spun fibers.

Ensuring the maximum draw ratio for each polymer at both stages involved fine control of the spinning speed to achieve stable spinning without breakage, to achieve highly oriented fibers. The schematic representation of the spinning line is given in Figure 1.

Using this setup, spools of monofilament fibers were produced in quantities sufficient for testing their mechanical properties.

#### 2.2.4. Fibers’ Mechanical Properties

The mechanical properties of the fibers were studied using an Instron 1122 tensile testing machine over a fiber length of 10 mm at a tensile rate of 10 mm·min^−1^. To obtain mechanical characteristics for each batch of fibers, the data were averaged over at least 10 samples according to the standard [40]. All measurements were performed at 23 ± 2 °C.

#### 2.2.5. Optical Analysis

All fiber samples were examined by optical microscopy before mechanical measurements began in order to determine the diameter and control the variation in the thickness and defects of the fiber (surface roughness; bean shape; the presence of vacuoles or voids; porosity, which can be seen by the opacity of the fiber).

For each series of fibers, each fiber sample was examined for mechanical measurements for variations in thickness and defects (at least 10 samples in the series). A monofilament of fiber placed in a tensile cell was placed under a microscope and the diameter was determined at three points. If there was a deviation of more than 10%, or if there were any defects, the sample was rejected.

The fiber diameter was determined using a Biomed 6PO microscope (Biomed, Moscow, Russia) equipped with a ToupTek E3ISPM5000 camera (ToupTek Photonics Co., Hangzhou, China). The measurement was carried out using a 20× objective in a bright field (calculated optical resolution 0.2 µm/pixel).

#### 2.2.6. Thermal Analysis

To determine the fraction of the residual solvent in the fiber and the released solvent-containing composition during spinning, the synchronous thermal analysis (STA) method on a STA 449 F1/F3 Jupiter (Netzsch, Selb, Germany) in air, with a heating rate of 10 K/min, was used. The airflow rate through the measuring cell was 50 mL·min^−1^, and the protective inert gas (Ar) flow rate was 20 mL·min^−1^.

STA measurements were carried out on film samples for better adherence to the bottom of the crucibles. Films of 40–100 µm in thickness were cast from solutions in DMSO, dried to a constant weight, and boiled in distilled water (electrical conductivity less than 1 µS·cm^−1^) three times for an hour and dried under a vacuum of 0.05 mbar at 80 °C within 4 h. The mass of the samples ranged from 4.5 to 5 mg.

The study of the carbon residue during the PAN carbonization process involved placing powder samples in alumina crucibles and subjecting them to oxidative stabilization in a DIL 402 (Netzsch, Selb, Germany) instrument under the following temperature–time regime: heating from 25 to 200 °C at a rate of 10 K·min^−1^, from 200 to 255 °C at 0.2 K·min^−1^, and an isothermal segment at 255 °C for 8 h. The airflow rate through the cell was 100 mL·min^−1^. The stabilized powder samples were heated in a nitrogen atmosphere up to 1500 °C at a rate of 10 K·min^−1^ (nitrogen flow through the cell at 100 mL·min^−1^). Cooling was uncontrolled from 1500 °C to 1000 °C for 10 min followed by gradual cooling to room temperature over 3 h. Mass losses due to specific temperature treatments were calculated by weighing the crucibles with powders after each experiment.

#### 2.2.7. FT-IR Spectroscopy

The IR spectra of all samples were obtained in transmission mode using the Fourier-transform infrared spectrometer Tensor 27 (Bruker, Billerica, MA, USA) (16 scans, resolution 2 cm^−1^, range 4000–600 cm^−1^). The IR spectra of the original polymers were obtained by measuring their films in transmission mode. After stabilization, the samples were ground with KBr in a mortar, pressed into tablets using a manual press, and measured in transmission mode.

## 3. Results and Discussion

Continuous fiber spinning from a PAN solution without using coagulation baths is realized by suppressing capillary instability before the onset of deformation-induced phase separation and requires two main conditions: a high degree of solution jet deformation, which promotes the stretching and orientation of polymer chains [41,42], and low polymer-to-solvent affinity [43]. This can be achieved by using either an initially “poor” solvent or by adjusting the air humidity during spinning, which promotes slight water uptake by the jet during stretching and initiates phase separation. In the case of PAN homopolymers, even in a good solvent, the possibility of achieving high polymer draw ratios is limited due to the formation of intermolecular associations between the nitrile groups themselves [44,45] or through solvent molecules [38]. Such processes are exacerbated with increasing polymer concentration, resulting in spontaneous gelation even in the presence of plasticizing monomers in the case of PAN copolymers [46] in concentrated solutions with a dense network of entanglements. Therefore, to successfully implement mechanotropic spinning from PAN homopolymers, the differences in the rheological behavior of solutions were analyzed and conditions for fiber formation were selected.

### 3.1. Solutions Rheology

The rheological behavior of PAN solutions with varying molecular weights was investigated in shear and oscillatory modes to determine the optimal concentration of the spinning solution. For the mechanotropic spinning of a continuous fiber with a uniform thickness, phase separation in the solution must occur before the onset of Plateau–Rayleigh capillary instability. Therefore, the solutions must possess sufficient viscosity while the initial stages of flow are not enable to control the fiber diameter [34,47]. All measurements were conducted at 70 °C to account for possible gelation, characteristic of concentrated PAN solutions due to strong nitrile–nitrile interactions [48]. The flow curves of the PAN solutions are presented in Figure 2.

All solutions exhibit a uniform behavior with a region of maximum zero viscosity (*η*_0_), followed by a decrease in viscosity due to reaching the critical shear rate at which the polymer in the solution does not have sufficient time for polymer relaxation [49]. With an increase in the molecular weight, a shift of the zero-viscosity boundary (shown in Figure 2a by an arrow) towards lower shear rates was observed, attributed to an increase in the time required for the polymer relaxation during the flow. Creating conditions for the stretching of polymer chains during the elongational flow of the solution can be achieved at deformation rates exceeding the characteristic relaxation time of the chains, according to the Weissenberg number (*Wi* >> 1) [50]. Accordingly, this condition can be met by increasing the deformation rate (external stretching) or increasing the relaxation time determined by the polymer concentration and its molecular weight [51,52]. Notably, as seen in the flow curves, the increase in molecular weight has a significantly greater impact on the position of the *η*_0_ boundary compared to the increase in polymer concentration.

For AN-1, the viscosity changes from 1 to ~700 Pa·s as the concentration increases from 15 wt% to 30 wt%, with the *η*_0_ boundary shifting from 100 to 1 s^−1^. In contrast, for a 15 wt% solution of AN-4, the deviation in *η*_0_ starts at 0.02 s^−1^ with a viscosity equal to the ~30 wt% solution of AN-1. The values of *η*_0_ are well correlated with the polymer’s molecular weight, as evident in the comparison of a series of 15 wt% solutions (Figure 2b).

Studying the viscoelastic properties of solutions in oscillation mode allows for determining the elastic and loss moduli over a wide frequency range and calculating the relaxation times of the solutions, providing the opportunity to select the spinning mode.

The frequency dependencies of the elastic and loss moduli, determined under conditions of linear viscoelasticity, are presented in Figure 3.

The obtained data exhibit typical characteristics of viscoelastic polymer solutions [53]. As expected, both viscosity and elasticity values increase with concentration. The increase in elasticity becomes more pronounced, and deviations from the Maxwell model become more significant as the molecular weight increases. In the investigated series of AN-1 solutions, the crossover (equality of elastic and loss moduli) is observed only in the concentrated 30 wt% solution at a high frequency of ~300 rad·s^−1^ (at moduli values ~4000 Pa). In a series of solutions with a higher-molecular-weight polymer, AN-2, the crossover at ~300 rad·s^−1^ is already observed in a 15 wt% solution with lower modulus values (~1000 Pa). Further increasing the molecular weight leads to a greater shift of the crossover towards lower frequencies and lower moduli for solutions with the same 15 wt% concentration: 10 rad·s^−1^ (400 Pa) and 2.3 rad·s^−1^ (300 Pa) for AN-3 and AN-4, respectively. Such changes indicate significant alterations in relaxation times in the studied systems.

To explore differences across the entire dataset, modulus dependencies were transformed into a frequency-invariant form (Cole–Cole dependence [54,55]). The results are presented in Figure 4.

In such coordinates, the increase in the difference between the elastic and loss moduli becomes visible as the polymer’s molecular weight increases, particularly pronounced in the low-stress region (low-frequency measurements). Such differences indicate a change in relaxation processes in the solutions as the molecular weight increases, as the number of entanglements per molecule significantly increases with molecular weight growth [56,57]. The increased storage modulus (*G*′) leads to deviations from the classical Maxwell viscoelastic fluid model (dashed line in Figure 4). At high concentrations and frequencies of impact (the region of maximum deviation from the Maxwell model), an approach towards highly elastic behavior is observed, characterized by a decrease in *G*″ at a constant value of *G*′ [58].

For a detailed analysis of the rheological state of the systems, the power-law dependencies of the storage modulus on frequency (*G*′~*ω^α^*) and the loss modulus on frequency (*G*″~*ω^β^*) were compared in the low-frequency region, approaching the “terminal zone”, corresponding to the limiting relaxation times of polymer chains. The viscoelastic state of the solution corresponds to the Maxwell model, where *α* = 2 and *β* = 1. Figure 5 presents the values of *α* and *β* for PAN solutions with different molecular weights as a function of concentration.

When examining the dependencies of the exponents on the concentration of the polymer (Figure 5a), a trend towards decreasing values of *α* and *β* is observed. The situation changes when using *c*[*η*]—a parameter characterizing the polymer’s volume in the solution as an argument. The use of *c*[*η*] revealed a common tendency of increasing deviation from the Maxwell model at low frequencies as the molecular weight of PAN increased.

The use of polymers with different molecular weights allowed us to study a wide range of concentrations. It was noted that in the *c*[*η*] range from 20 to 45, the values of *β* remained constant, and *α* slightly decreased. Then, the character of the dependencies qualitatively changed, likely associated with the transition to the region of concentrated solutions. A linear reduction in the values of *α* and *β* with a steeper slope was observed, indicating an increase in the solution structuring. Thus, the degree of deviation from the Maxwell model depends solely on the volume occupied by the polymer coil.

For the implementation of mechanotropic spinning, a key factor is the relaxation time, determining the stretching ratio required for the phase separation during uniaxial deformation [34,41,59]. Figure 6 represents the dependence of the maximum relaxation time at 0.1 rad·s^−1^, as the closest to the macromolecule’s relaxation time *θ_m_*, and the characteristic relaxation time when elasticity begins to dominate over viscosity *θ_c_*, determined by the following formulas [51,58]:
(1)$$\begin{array}{c} \theta_{m}=\frac{G^{\prime }}{G^{\prime \prime }\omega } \end{array}$$
(2)$$\begin{array}{c} \theta_{c}=\frac{1}{\omega_{cr}}, \end{array}$$where $\omega_{cr}$ is the crossover frequency determined by the equality of the elastic and loss moduli.

The relaxation of macromolecules requires significantly more time. Thus, the dependence of *θ_m_* is located consistently higher than the dependence of *θ_c_* over the entire range and the second one is characterized by a steeper slope. The obtained data indicate the existence of unified, linear semi-logarithmic coordinates, describing the relaxation behavior of macromolecules at different frequencies. Such a relationship suggests a range of deformation rates (frequencies) for polymer solutions, necessary to achieve conditions for stretching the chains before the onset of phase separation, for solutions with different molecular weights and concentrations expressed through the parameter *c*[*η*].

### 3.2. Fiber Spinning

The rheology analysis of the solutions presented above allowed for the selection of optimal concentrations and spinneret temperature corresponding to the solution temperature for continuous fiber spinning using the mechanotropic method. The spinning was carried out at maximum drawing ratios at each stage to determine the stretching potential of the homopolymers series obtained by the anionic polymerization. Table 2 presents the spinning regime parameters.

During the fiber spinning, the amount and composition of the liquid released from the as-spun fiber were tested. The compositions of the fiber before starting the orientational drawing and of the final fiber were also determined.

At first, it was necessary to determine whether this liquid was a solution or a solvent. For this purpose, a series of preliminary experiments were carried out to construct a correlation dependence of the heat flow on the PAN content (Figure 7a).

A series of solutions with a PAN content of 0.1, 0.5, and 1 wt% in DMSO were prepared. They were placed in DSC crucibles and dried to a constant weight. Then, the thermal effect from the cyclization of PAN upon heating to 400 °C was measured and a calibration curve was constructed. Finally, samples of the released and collected solvents were analyzed to determine the concentration of PAN based on the thermal effect. It was found that the liquid contained 0.2 wt% PAN.

Thus, it can be concluded that a small amount of PAN is washed out of the solution jet along with the solvent. Presumably, this may be due to the process of incomplete amorphous phase separation [60], when the system, upon crossing the binodal point, separates into two phases, one of which is enriched with the polymer, and the other is enriched with the solvent (dilute solution).

In the second stage, the composition of the liquid collected from the fiber was determined by optical refractometry using a calibration dependence for the DMSO–water system. The liquid was found to consist of 80 wt% DMSO and 20 wt% water. The data are shown in Figure 7b. That analysis showed that already at the first stage, without using coagulants, it is possible to collect about 80 wt% of the solvent, which consists of 80 wt% DMSO, which significantly simplifies its subsequent recovery for regeneration.

Finally, the as-spun and final fibers were studied by TG analysis upon heating to 400 °C. The characteristic steps were used to estimate the content of water and DMSO in the fiber after phase separation and active release of the solvent (first stage) and after all stages of drawing and drying. The data are shown in Figure 7c. The TG curve of the as-spun fiber shows two steps corresponding to the maximum rate of evaporation of water (at 80 °C) and DMSO (130 °C). It is shown the freshly formed fiber contains ~17 wt% DMSO and ~8 wt% water, which is probably adsorbed by the fiber during drawing in air and winding onto a roller due to the high hydrophilicity of the solvent. The presence of the liquid phase in fibers significantly simplifies and facilitates the following stage of plasticizing drawing, allowing us to achieve a high degree of molecular orientation in the fiber from the homopolymer. At the same time, after the stages of washing, thermal drawing, and drying, as can be seen in Figure 7c, the residual solvent is removed (the content of DMSO and water is less than 0.2 wt%).

The surface morphology of fibers formed in similar spinning conditions is shown in Figure 8. Images were obtained by optical microscopy in transmission mode.

The obtained fibers were transparent, exhibiting no visible defects. No differences in morphology were found among the copolymers.

The results of mechanical testing (Figure 9) reveal that the highest tensile strength was observed in the AN-4 sample, i.e., the sample with the highest molecular weight. However, in the case of the elastic modulus and elongation at break, a direct correlation with molecular weight was not observed.

It is expected that mechanical properties correlate well with the ultimate draw ratio, and the highest draw ratio was achieved when using the highest-molecular-weight polymer with the lowest concentration of the spinning solution (the dependence of strength on molecular weight can be seen in Figure 9a). In this case, well-oriented high-strength fibers were obtained with mechanical property values comparable to those of ternary PAN copolymers [35,53]. The use of a less viscous and more elastic solution likely contributes to better chain stretching during elongation. A similar improvement in properties with decreasing concentration was observed in the formation of fibers from ultra-high-molecular-weight polyethylene [61].

### 3.3. Thermal Properties

The thermal behavior of linear PAN in the air is presented in Figure 10. In the examined temperature range, PAN samples exhibit a single exothermic peak, corresponding to the cyclization reaction of nitrile groups [27,32,33,39,62,63]. The peak location is slightly different for the samples and equals 292 °C for samples AN-1 and AN-3 and 296 °C for sample AN-4, which can be explained by film size differences or different contacts of the polymer with the crucible bottom.

The onset temperature of intensive decomposition (“step” in mass loss) for all samples occurs at ~296 °C (Figure 10b). The weight loss for all samples obtained by anionic polymerization, after heating to 500 °C, is 24–25 wt%, which is less than that of the reference sample (27 wt%).

To determine the carbon residue, polymer samples were thermally stabilized in the air in accordance with the procedure described in the Experimental section. IR spectra of the samples were obtained to confirm complete cyclization, as evidenced by the disappearance of the characteristic moderately intense stretching vibrations at 2243 cm^−1^, indicating the disappearance of the nitrile group (Figure 11).

It can be observed that the spectra of samples AN-2, AN-3, and AN-4 differ from AN-1 and the reference sample. The presence of broad peaks at 1615 and 1580 cm^−1^ is evident, indicating the presence of stretching and deformation oscillations of the aromatic ring, as well as the presence of amine groups in this region (aliphatic amines in the 1150 and 1250 cm^−1^ range).

The weight losses of polymers obtained by anionic polymerization and the reference homopolymer sample after thermo-oxidative stabilization and carbonization in a nitrogen atmosphere up to 1500 °C are presented in Figure 12.

It can be seen that the differences in the weight loss of samples after thermo-oxidative stabilization and carbonization are similar, indicating that the variations between the samples occur precisely at the thermo-oxidative stabilization stage. After carbonization to 1500 °C, the cox residue is around 48 wt%, which is a high value. The reference sample loses approximately 15 wt% more compared to samples AN-2–AN-4, which may be attributed to the synthesis peculiarities. The obtained data qualitatively coincide with the STA data of the film samples.

Despite the identical picture of thermal behavior (integral area, position and intensity of exothermic effects, mass loss in the area of intensive decomposition) when heating the polymers in question in an air atmosphere, the results obtained during stabilization and subsequent high-temperature treatment differ strongly: in the process of stabilization at a temperature of 255° C, a PAN-C loses almost twice as much weight, and during the subsequent high-temperature treatment of stabilized samples (which are confirmed by the results of IR spectroscopy: there are no absorption bands responsible for nitrile groups in the spectra, which indicates the sample has undergone oxidative stabilization ), its weight loss is only comparable with the lowest-molecular-weight sample (the molecular weight of which is two times less than the commercial sample). The use of samples obtained by an anionic method with a molecular weight comparable to and greater than the molecular weight of a commercial polymer obtained by radical polymerization allows us to significantly increase the carbon yield in the considered temperature range.

## 4. Conclusions

In this study, we demonstrate the feasibility of producing high-strength fibers from homopolymers of PAN synthesized using the anionic polymerization method without metal-containing initiators. An analysis of the rheological behavior of concentrated solutions has allowed us to identify a range of deformation rates for polymer solutions necessary to achieve conditions for stretching the chains before the onset of phase separation in the course of mechanotropic spinning. Unifying the key conditions for PAN solutions with different concentrations of polymers with different molecular weights is expressed via the parameter *c*[*η*].

The fiber formation under these conditions, using the mechanotropic method at the maximum draw ratio for each sample, facilitated an in-depth exploration of the potential of each specimen. It was observed that the highest draw ratio, and consequently, fiber strength, was attained with the less-concentrated PAN solutions possessing the highest molecular weight and exhibiting the longest relaxation times.

It was shown that, at the first stage, it is possible to collect a solvent containing about 80 wt% DMSO, which significantly facilitates its subsequent recovery and return to the process. The as-spun fiber contains about 20 wt% residual DMSO and 10 wt% water, which act as plasticizers. This allows us to achieve a high degree of orientational drawing of the fiber and to obtain high-strength fibers even from PAN homopolymers.

Consequently, high-strength fibers with a diameter of 12 µm, a strength of up to 800 MPa, an elastic modulus up to 10 GPa, and an elongation at break of 20% were obtained.

Furthermore, an investigation into the thermal behavior of powder and film samples of the polymer revealed the anticipated invariance in the kinetics of thermo-oxidative stabilization and carbon residue during carbonization. This residue constituted 47 wt%, which is 15 wt% higher than that of the industrial homopolymer of PAN synthesized by radical polymerization, selected as a reference sample.

In the continuation of this work, it is planned to obtain a sufficient amount of fiber to test detailed regimes of thermal–oxidative stabilization and carbonization to obtain high-strength carbon fibers with increased coke residue and compare them with traditional PAN copolymers, as well as obtain CF fibers from copolymers obtained by the anionic method.

## Figures and Tables

**Figure 1 polymers-16-01185-f001:**
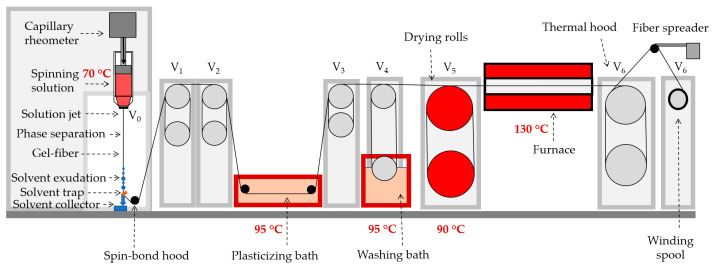
Scheme of the mechanotropic spinning line. *V*_0_ is the linear flowing speed from the spinneret, and *V*_1_* − V*_6_ are the winding speeds.

**Figure 2 polymers-16-01185-f002:**
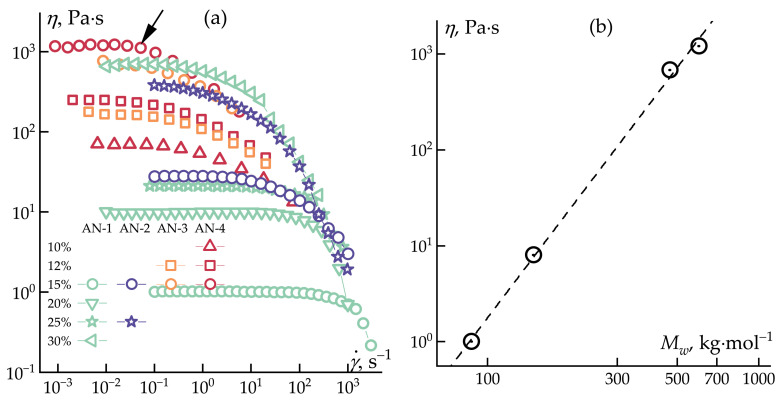
Flow curves for PAN solutions with different concentrations and molecular weights at 70 °C (**a**), and dependence of maximum zero viscosity of 15 wt% solutions on molecular weight (**b**).

**Figure 3 polymers-16-01185-f003:**
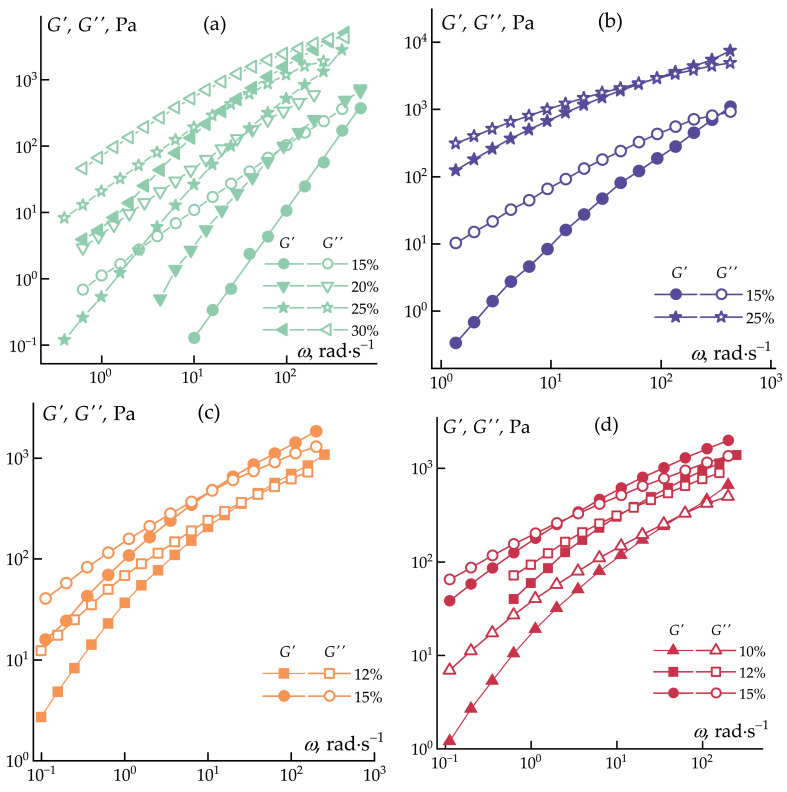
Frequency dependencies of elastic (*G*’) and loss (*G*″) moduli for PAN solutions over a wide concentration range at 70 °C, for AN-1 (**a**), AN-2 (**b**), AN-3 (**c**), and AN-4 (**d**) homopolymers.

**Figure 4 polymers-16-01185-f004:**
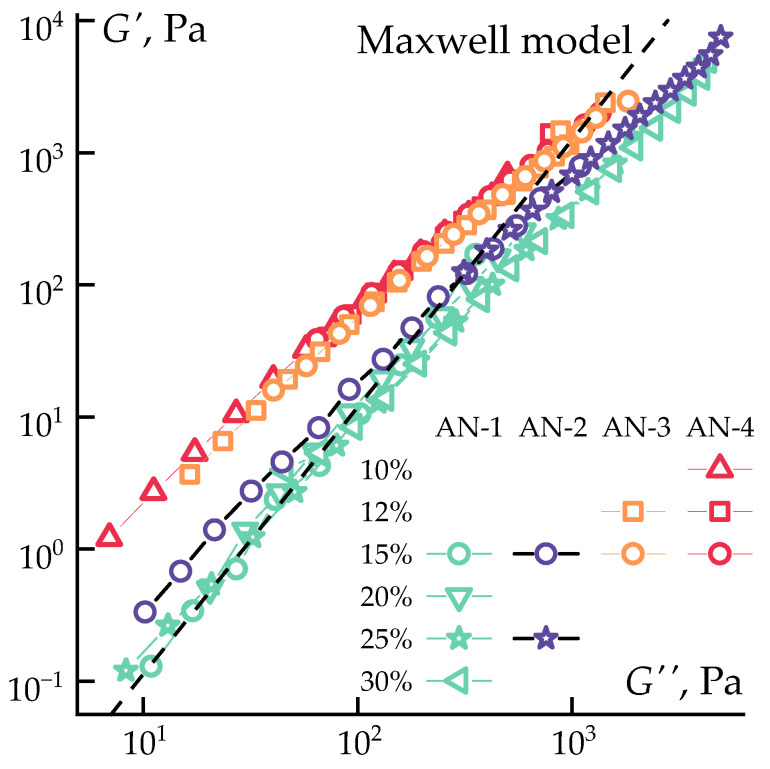
Cole–Cole plot for PAN solutions with varying molecular weights and concentrations at 70 °C.

**Figure 5 polymers-16-01185-f005:**
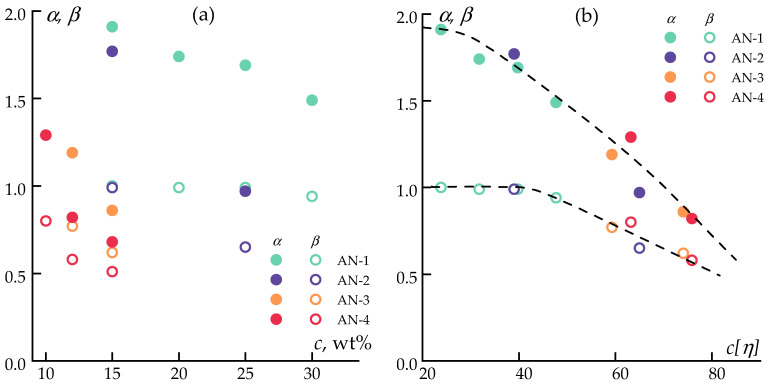
Concentration dependencies of the exponents in power law for the storage (*α*) and loss (*β*) moduli on frequency (**a**), and on the volume occupied by the macromolecule (*c*[*η*]) (**b**) at 70 °C.

**Figure 6 polymers-16-01185-f006:**
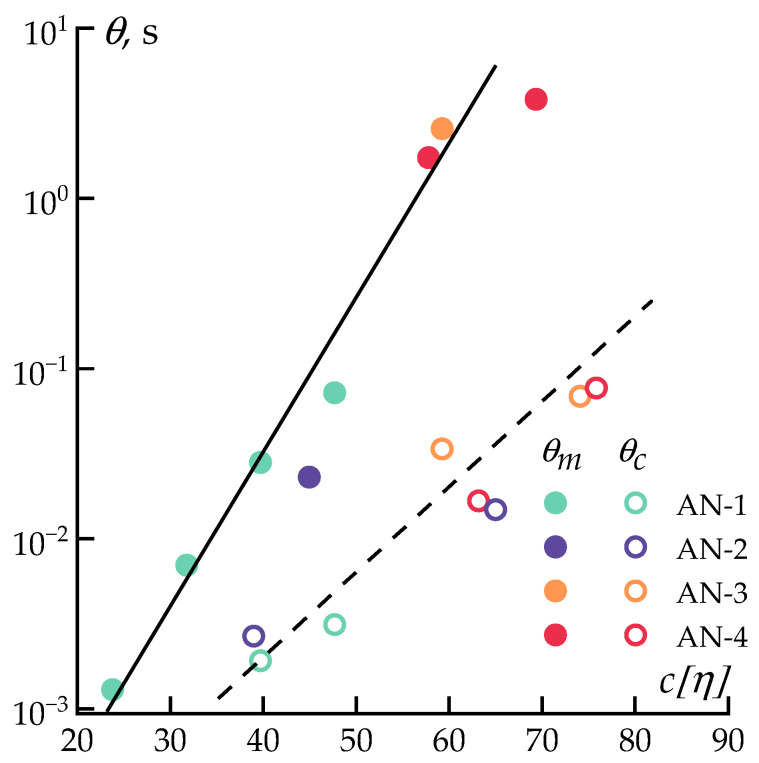
Dependencies of the maximum (solid symbols) and characteristic (open symbols) relaxation times (*θ*) on macromolecular volume (*c*[*η*]) for various polymers at 70 °C.

**Figure 7 polymers-16-01185-f007:**
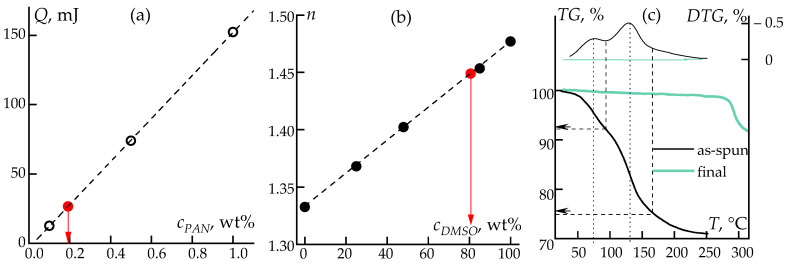
Correlation of the thermal effect (*Q*) of PAN on the concentration of the solution (**a**), and the refractive index (*n*) of the DMSO–water mixture: black dots indicate calibration data, red dots indicate the value of experimental data (**b**). TG and DTG data of as-spun and final fibers. The short dashes show the maximum rate of the mass change, the dashes indicate quasi-end of the evaporation stages (**c**).

**Figure 8 polymers-16-01185-f008:**
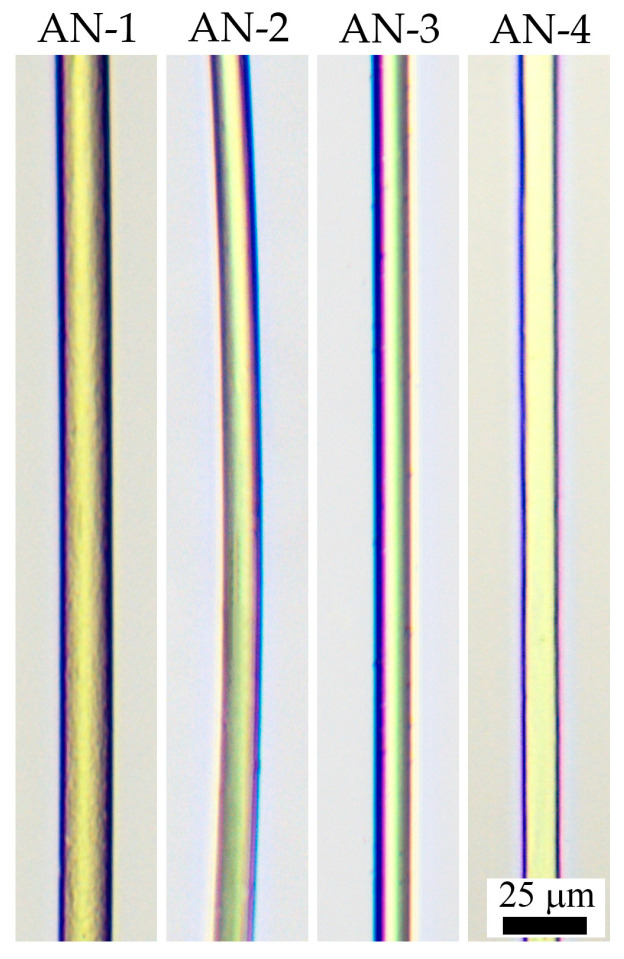
Images of the obtained PAN fibers.

**Figure 9 polymers-16-01185-f009:**
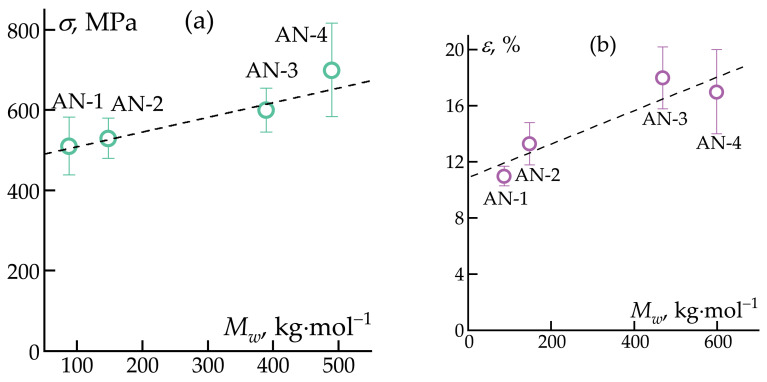
Correlations of tensile strength (**a**), elongation at break (**b**), modulus of elasticity (**c**), and diameters (**d**) with *M_w_* for PAN fibers.

**Figure 10 polymers-16-01185-f010:**
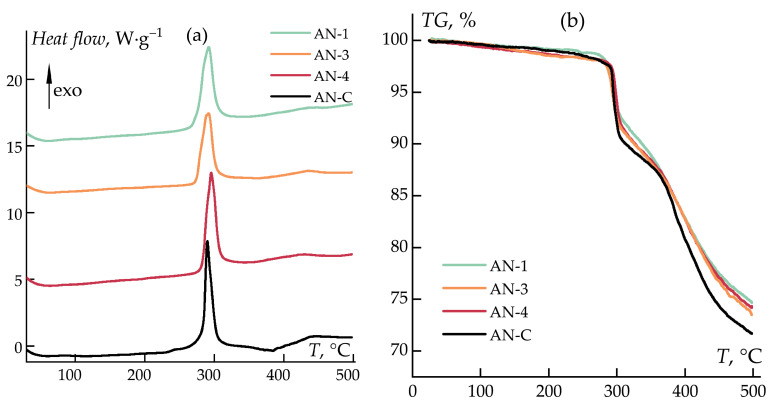
DSC (**a**) and TG (**b**) of homopolymers AN-1, AN-3, and AN-4, and the reference sample. Heating rate: 10 K/min, air.

**Figure 11 polymers-16-01185-f011:**
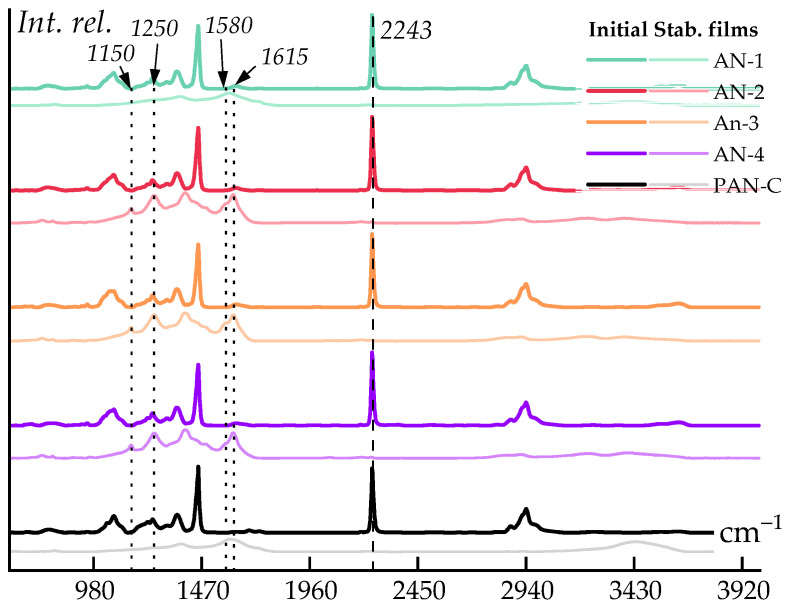
IR spectra of the original and thermally oxidatively stabilized films of PAN samples.

**Figure 12 polymers-16-01185-f012:**
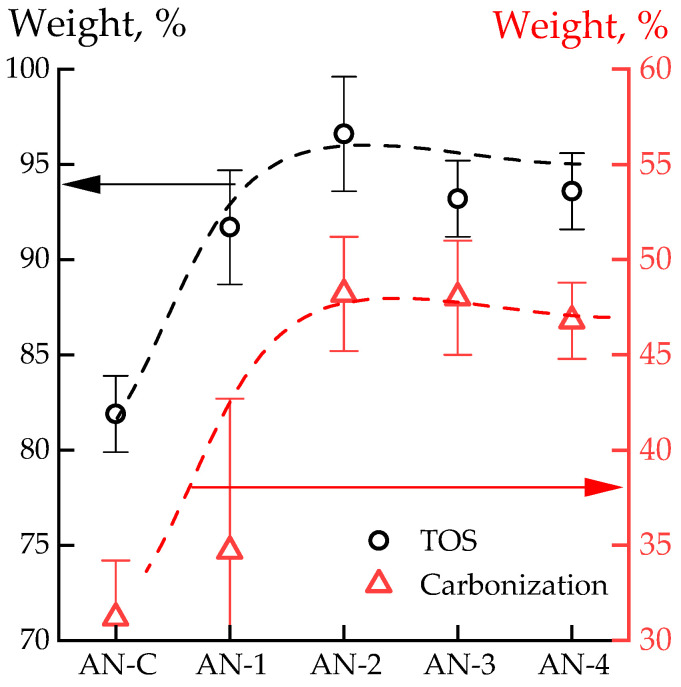
Weight loss data of PAN powders after TOS in air (at 255 °C) and subsequent carbonization in N_2_ (to 1500 °C).

**Table 1 polymers-16-01185-t001:** The characteristics of the polymers used in the study.

Sample	*M_n_*, kg·mol^−1^	*M_w_*, kg·mol^−1^	*Ð* ^1^	[*η*], dL·g^−1^	*M_η_*, kg·mol^−1^
AN-1	40	87	2.2	1.56	83
AN-2	82	148	1.8	2.7	171
AN-3	217	469	2.2	5.06	434
AN-4	344	599	1.7	6.32	599
AN-C [38]	88	150	1.7	1.75	88

^1^ *Ð* is the polydispersity index.

**Table 2 polymers-16-01185-t002:** Parameters of the mechanotropic spinning.

Sample	*c*, wt%	*T_s_* ^1^, °C	*V*_0_ m·min^−1^	Roller’s Velocity, m·min^−1^						Drawing Ratio (*V*_6_/*V*_1_)	Total Drawing Ratio (*V*_6_/*V*_0_)
				*V* _1_	*V* _2_	*V* _3_	*V* _4_	*V* _5_	*V* _6_		
AN-1	30	25	0.16	1.3	1.4	2.4	6	6.3	12	9.2	75
AN-2	25	90	0.04	2.2	2.6	3.7	5.5	5.6	10.3	4.7	258
AN-3	13.5	70	0.1	4.2	4.8	16	16.8	16.8	24.5	5.8	245
AN-4	12	25	0.08	4.4	5	12.9	16.3	16.6	44	10	550

^1^ *T_s_* is the temperature of the spinneret.

## Data Availability

The data that support the findings of this study are available from the corresponding author upon reasonable request. The data are not publicly available due to privacy.

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
