# Peer review of "Non-Coagulant Spinning of High-Strength Fibers from Homopolymer Polyacrylonitrile Synthesized via Anionic Polymerisation"

_polymers, 2024, doi:10.3390/polym16091185_

Round 1
Reviewer 1 Report
Comments and Suggestions for Authors
The manuscript titled "Non-coagulant spinning of high-strength fibers from homоpolymer polyacrylonitrile synthezed via anionic polymerisation" needs revision in the following aspects:
1. Abstract has to be included with the need for the current experimental study.
2. Almost 50 % of the references are way too old. Authors may kindly replace them with recent references.
3. The thermal stability and crystallization behaviour of the fibers need in-depth discussion. The current discussion is not sufficient to understand their behaviour.
4. What is the reason for the slight drop in the modulus of elasticity of AN-4? Kindly explain.
5. Limitations of the current study and the prospects of future study can be included in the conclusion section.
Comments on the Quality of English LanguageThe English language needs minor finetuning.
Author Response
Answers for reviewers
Reviewer 1
First of all, we are grateful to Reviewer for the thorough revision of our work. We believe it helped to improve its quality. We appreciate the time and effort you have put into providing feedback to help us improve our work. Answers are marked in purple in the attached file; all corrections in the article were made in editing mode. Please find our responses to your comments below:
The manuscript titled "Non-coagulant spinning of high-strength fibers from homоpolymer polyacrylonitrile synthezed via anionic polymerisation" needs revision in the following aspects:
- Abstract has to be included with the need for the current experimental study.
Answer: The Abstract has been rewritten:
The rheological properties, spinnability, and thermal-oxidative stabilization of high-molecular-weight linear polyacrylonitrile (PAN) homopolymers (molecular weights Mη = 90–500 kg/mol), synthesized via a novel metal-free anionic polymerization method, were investigated to reduce coagulant use, enable solvent recycling, and increase the carbon yield of the resulting carbon fibers. This approach enabled the application of the mechanotropic (non-coagulating) spinning method for homopolymers PAN solutions in a wide range of molecular weights and demonstrated the possibility of achieving a high degree of fiber orientation and reasonable mechanical properties. Rheological analysis revealed a significant increase in solution elasticity (G') with increasing molecular weight, facilitating the choice of optimal deformation rates for effective chain stretching prior to strain-induced phase separation during the eco-friendly spinning of concentrated solutions without using coagulation baths. The possibility of collecting ~80 wt% of the solvent at the first stage of spinning from the as-spun fibers was shown. Transparent, defect-free fibers with a tensile strength up to 800 MPa and elongation at break of about 20% were spun. Thermal treatment up to 1500 °C yielded carbon fibers with a carbon residue of ~50 wt%, in contrast to ~35 wt% for industrial radically polymerized PAN carbonized under the same conditions.
- Almost 50 % of the references are way too old. Authors may kindly replace them with recent references.
Answer: We agree with the Reviewer about the References should be up in date. However, the references list contains only 16 of 62 refs older 2010, mostly primary sources, subsequent works refer to them. With all due respect to the Reviewer, most fundamental laws and experimental results do not have a statute of limitations, so using only cross-reference links from fresh sources (for example, for the last 5 years) may not only be disrespectful to the original source, but may also introduce errors in the interpretation of certain results.
- The thermal stability and crystallization behaviour of the fibers need in-depth discussion. The current discussion is not sufficient to understand their behaviour.
Answer: Atactic PAN is not a crystalline polymer, so we did not investigate this.
The following text has been added to the article:
Despite the identical picture of thermal behavior (integral area, position and intensity of exothermic effects, mass loss in the area of intensive decomposition) when heating the polymers in question in an air atmosphere, the results obtained during stabilization and subsequent high-temperature treatment differ strongly: in the process of stabilization at a temperature of 255° C, a PAN-C loses almost twice as much weight, and during subsequent high-temperature treatment of stabilized samples (which is confirmed by the results of IR spectroscopy: there are no absorption bands responsible for nitrile groups in the spectra, which indicates a deep stabilization process), its weight loss is only comparable with the lowest molecular weight sample (the molecular weight of which is two times less than the commercial sample). The use of samples obtained by anionic method with a molecular weight comparable to and greater than the molecular weight of a commercial polymer obtained by radical polymerization allows us to significantly increase the carbon yield in the considered temperature range.
- What is the reason for the slight drop in the modulus of elasticity of AN-4? Kindly explain.
Answer: We consider small deviations in the elastic modulus values to be due to measurement errors. In addition, it can be assumed that the sample with the highest molecular weight requires an additional stage of thermal drawing or plasticizing steam thermal drawing for better orientation of the chains.
- Limitations of the current study and the prospects of future study can be included in the conclusion section.
The following text was inserted into the article:
In continuation of the work, it is planned to obtain a sufficient amount of fiber to test detailed regimes of thermal-oxidative stabilization and carbonization to obtain high-strength carbon fibers with increased coke residue and comparison with traditional PAN copolymers, as well as obtaining СF-fibers from copolymers obtained by the anionic method.

Reviewer 2 Report
Comments and Suggestions for Authors
The topic of the paper is very interesting, but the manuscript needs to be completed and written more clearly for the reader.
Abstract:
The abstract states that the fibers have reasonable mechanical properties in one place and excellent mechanical properties in another.
Notes referring to the entire manuscript:
The nomenclature of the symbols used, their explanations and the units used should be clearly emphasized - both in the text and in the description of all figures and tables (e.g. Table 1. the meaning of Đ is not given...)
2.2.1 The polymer concentration used should be clearly defined.
2.2.3 It should be noted that these are monofilament fibers.
2.2.4 When defining the mechanical properties of fibers, it is necessary to define the number of individual measurements and the standard according to which the measurement was carried out.
It is necessary to clarify the purpose and methodology of the analysis performed with an optical microscope (number of measurements, method, magnification...)
I suggest to specify the fineness of the fibers.
3. I suggest moving the text in lines 191-205 to chapter 2.2.3.
Figure 8. shows more significant differences in diameter.
Author Response
Answers for reviewers
Reviewer 2
The topic of the paper is very interesting, but the manuscript needs to be completed and written more clearly for the reader.
The authors are grateful to the Reviewer for carefully reviewing the manuscript and making very valuable comments. All of them have been taken into account and appropriate changes have been made to the text. Answers are marked in purple in the attached file; all corrections in the article were made in editing mode.
Abstract:
The abstract states that the fibers have reasonable mechanical properties in one place and excellent mechanical properties in another.
Answer: The redundant phrase has been removed.
Notes referring to the entire manuscript:
The nomenclature of the symbols used, their explanations and the units used should be clearly emphasized - both in the text and in the description of all figures and tables (e.g. Table 1. the meaning of Đ is not given...)
Answer: The text has been proofread, all symbols were checked and emphasized.
2.2.1 The polymer concentration used should be clearly defined.
Answer: Through the Manuscript the weight concentrations were used. Now the % symbols have been changed on wt%.
2.2.3 It should be noted that these are monofilament fibers.
Answer: Yes, it is. We used a monofilament spinneret in this study. The information about monofilament fibers has been added to clarify the text:
“Using this setup, spools of monofilament fiber were produced in quantities sufficient to test their mechanical properties.”
2.2.4 When defining the mechanical properties of fibers, it is necessary to define the number of individual measurements and the standard according to which the measurement was carried out.
Answer: The section states that 10 samples per batch were used. Additional information and standards have been added.
It is necessary to clarify the purpose and methodology of the analysis performed with an optical microscope (number of measurements, method, magnification...)
Answer: We agree with the Reviewer, the optical microscopy was carried out to solve the following tasks:
1) Determination of fiber diameter for mechanical tests;
2) determination of fiber thickness variations;
3) determination of the presence or absence of surface defects (roughness; bean-like shape, which can be seen by the flattening of the fiber; the presence of vacuoles or voids; porosity, which can be seen by the opacity of the fiber).
The following text has been added to the article:
All fiber samples were examined by optical microscopy before mechanical measurements began in order to determine the diameter, control the variation in thickness and defects of the fiber (surface roughness; bean-shaped shape, the presence of vacuoles or voids; porosity, which can be seen by the opacity of the fiber) was carried out.
For each series of fibers, each fiber sample was examined for mechanical measurements for variations in thickness and defects (At least 10 samples in the series). A monofilament of fiber placed in a tensile cell was placed under a microscope and the diameter was determined at three points. If there was a deviation of more than 10%, or if there were any defects, the sample was rejected.
The measurement was carried out using a 20x objective in bright field (calculated optical resolution 0.2 µm/pixel).
I suggest to specify the fineness of the fibers.
Answer: This is really important information. The fibers diameter was shown in the manuscript, but now we have added more detail data of diameters (Figure 9(d))
- I suggest moving the text in lines 191-205 to chapter 2.2.3.
Answer: With all due respect to the Reviewer, we consider it important to leave this text in the current section, because it explains the need for the following rheological studies to select the fiber spinning concentration.
Figure 8. shows more significant differences in diameter.
Answer: Thank you for your valuable comment, it turned out that an error was made when applying the scale bar and combining the images.

Round 2
Reviewer 1 Report
Comments and Suggestions for Authors
The revised manuscript is improved as per the reviewer's comments and suggestions. Now the manuscript can be accepted in is present form. Best wishes to all the authors.
Reviewer 2 Report
Comments and Suggestions for Authors
-